# Isavuconazole-Amphotericin B and Isavuconazole-Caspofungin In Vitro Synergic Activity Against Invasive Pulmonary Aspergillosis Molds Isolates

**DOI:** 10.3390/antibiotics14100993

**Published:** 2025-10-04

**Authors:** Maddalena Calvo, Michelangelo Caruso, Adriana Antonina Tempesta, Laura Trovato

**Affiliations:** 1U.O.C. Laboratory Analysis Unit, A.O.U. Policlinico “G. Rodolico-San Marco” Catania, 95123 Catania, Italy; maddalenacalvo@gmail.com; 2Department of Biomedical and Biotechnological Sciences, University of Catania, 95123 Catania, Italy; miche74caru@gmail.com (M.C.); adriana110995@gmail.com (A.A.T.)

**Keywords:** invasive pulmonary aspergillosis, isavuconazole, caspofungin, amphotericin B, *Aspergillus* spp., synergy testing

## Abstract

**Background/Objectives**: Invasive pulmonary aspergillosis (IPA) represents a critical respiratory condition mainly caused by *Aspergillus fumigatus* and other ubiquitous species such as *Aspergillus flavus*, *Aspergillus niger*, and *Aspergillus terreus*. IPA clinical management has been complicated by diagnostic challenges and therapeutic difficulties due to antifungal intrinsic or secondary resistance episodes. Despite this assumption, few scientific data have been reported about possible antifungal drug combinations. Herein, we propose an experimental evaluation using isavuconazole/amphotericin B and isavuconazole/caspofungin in vitro synergy assays to investigate their combined activity on *Aspergillus* spp. IPA clinical isolates. **Methods**: We globally analyzed 55 *Aspergillus* spp. isolates, practicing the gradient test methods with single and combined antifungal drugs through the MIC Strip test (Liofilchem, Roseto degli Abruzzi, Italy). The collected MIC values were interpreted according to the EUCAST guidelines and classified as synergy, additivity, indifference, and antagonism cases through a FIC index calculation. A statistical analysis on species’ correlation with particular synergy testing results was finally provided. **Results**: Despite an interesting activity against *A. fumigatus*, isavuconazole/amphotericin B did not report statistical significance, obtaining a consistent antagonism percentage (43.6%). On the other hand, isavuconazole/caspofungin showed a promising in vitro synergic activity, except for *A. flavus* isolates. **Conclusions**: Synergy testing demonstrated a significant species-specific trend. Future studies should be focused on *Aspergillus* spp. isolates and antifungal in vitro synergy testing, aiming to discourage or recommend any specific antifungal combinations, depending on the isolated species.

## 1. Introduction

Invasive pulmonary aspergillosis currently represents a significant clinical challenge due to difficult microbiological workflows and frequent diagnostic delays [1,2,3].

Fungal culture remains the gold standard method to isolate *Aspergillus* species and provide complete antifungal susceptibility testing [4,5]. However, serological assays for galactomannan antigen detection and molecular tests may integrate imaging and clinical symptoms to define the aspergillosis diagnosis [6]. Recent antifungal resistance has enhanced the importance of always identifying and testing the antimicrobial susceptibility of fungal isolates. Aspergillosis therapy includes triazoles, amphotericin B, and echinocandins. Specifically, voriconazole, isavuconazole, and posaconazole are first-line agents in the case of pulmonary aspergillosis. Unfortunately, *Aspergillus* spp. isolates may express mutations in the sterol-demethylase gene *cyp51A*, reporting a reduction in the affinity between azole drugs and their targets [7]. Furthermore, the same isolates occasionally reveal efflux pumps overexpression, with a consequent reduction in azole concentrations within the fungal cell. Azole resistance generally depends on extensive agricultural azole usage or prolonged azole therapies within specific hospital settings [7]. The progressive azole resistance detection led to the intensification of amphotericin B insertion among aspergillosis therapeutic choices. In vitro evaluations demonstrated an intrinsic amphotericin B resistance among *A. terreus* isolates, even if the specific resistance mechanism still remains unclear. Moreover, the increase in amphotericin B prescription caused higher minimum inhibitory concentration (MIC) values to be detected among other *Aspergillus* species [7]. Echinocandin resistance depends on the glucan-synthase codifying genes *FKS1* and *FKS2*, but it has rarely been detected among *Aspergillus* spp. isolates.

Refractory aspergillosis episodes have sometimes been treated through antifungal combination therapies, including azoles and echinocandins or azoles and amphotericin B [8,9]. The European Committee on Antimicrobial Susceptibility Testing (EUCAST) and the Clinical and Laboratory Standards Institute (CLSI) established the criteria to standardize antifungal susceptibility testing, reporting epidemiological cut-offs (ECOFFs) in the absence of clinical breakpoints (BPs). According to the same guidelines, the gradient test method may be performed to classify azoles, echinocandins, and amphotericin B MIC values for *Aspergillus* spp. The above-mentioned methods can also be used to test the in vitro synergic activity [10,11]. Recent literature data documented a potential in vitro synergy of isavuconazole and amphotericin B against different fungal isolates, including *Aspergillus* spp. isolates [12,13]. Additionally, Raffetin et al. [8] demonstrated similar results for caspofungin and isavuconazole combinations.

Herein, we propose an experimental in vitro study to address the challenge of identifying optimal antifungal combination therapy for invasive pulmonary aspergillosis, given increasing resistance among *Aspergillus* species and limited treatment options. We specifically investigate the synergic effects of isavuconazole-amphotericin B and isavuconazole-caspofungin on clinical isolates from invasive pulmonary aspergillosis episodes. Our evaluation aims to clarify which antifungal combinations may offer enhanced efficacy, depending on the isolated *Aspergillus* species. Moreover, it has the purpose of enriching literature data and general knowledge about antifungal combinations against molds.

## 2. Results

The experimental analysis globally collected 55 *Aspergillus* spp. isolates from respiratory samples such as bronchoalveolar lavage fluids, bronchial aspirates, and sputum samples. Figure 1 summarizes the *Aspergillus* spp. isolates distribution within the included hospital settings. According to our data, the Intensive Care Unit registered the highest *Aspergillus* spp. strains isolation rate (34.5%). Furthermore, this ward was the only one to report all the four main species (*A. fumigatus*, *A. flavus*, *A. terreus*, and *A. niger*), along with the Infectious Disease Unit, which showed a 21.8% isolation percentage. Pneumology revealed an 18.2% isolation rate, while 9.1% of the strains belonged to the Hematology Unit. Additionally, lower percentages emerged from the Emergency Room (5.4%) and the Internal Medicine Unit (3.63%). Finally, the Surgery, Oncology, and Transplant Units revealed the same isolation rates (1.8%).

*Aspergillus fumigatus* sensu strictu was the most isolated fungal species (40%), followed by *A. flavus* sensu strictu (25.4%), *A. niger* sensu strictu (20%), and *A. terreus* sensu strictu (14.5%). Table 1 provides details about species, hospital wards, biological samples, and MIC values for all the tested antifungal drugs. Additionally, Table 2 indicates the isolates number for each species, along with the total isolates number. Furthermore, the same table illustrates the obtained MIC ranges and MIC90 for all the collected species, reporting the eventual probable resistance in the case of MIC values higher than clinical breakpoints, epidemiological cut-offs, or the latest tested antifungal concentrations. *A. terreus* demonstrated the highest amphotericin B resistance rate (87.5%) due to its intrinsic resistance to this antifungal drug. Moreover, the same species showed the highest caspofungin resistance percentage (62.5%).

On the other hand, *A. fumigatus* and *A. niger* rarely showed amphotericin B or caspofungin resistance. *A. flavus* did not express caspofungin resistance, but it reported a single case (7.1%) of amphotericin B resistance. Finally, isavuconazole MIC values were never elevated for the tested *Aspergillus* spp. isolates.

Table 3 summarizes the in vitro gathered results, including isavuconazole and amphotericin B synergy testing, together with statistical significance indications. Overall, this antifungal combination reported additivity (18.2%) and synergy (7.8%) cases, along with a consistent antagonism percentage (43.6%) and 30.9% of indifference episodes. As regards the species-specific results, *A. fumigatus* revealed the highest additivity (27.7%) rate, while *A. flavus* reported the highest antagonism percentage (71.4%). *A. terreus* interestingly revealed the most elevated indifference rate (50.0%). Finally, synergy percentages were similar for all the tested species. None of the cases showed a statistical significance between the identified species and the synergy assay result.

Table 4 indicates the in vitro obtained results, including isavuconazole and caspofungin synergy testing, along with the statistical significance *p*-values. The total percentages document the same synergy rate (7.8%) as the previous antifungal combination, while the additivity episodes slightly increased (27.3). Antagonism cases (14.8%) appeared to be less numerous, whereas indifference percentage increased (43.6%) when compared to the other tested combinations. Remarkably, the comparison between the two tested antifungal combinations revealed a statistical significance (*p* < 0.0008) only with regard to antagonism, whose percentages were significantly higher in the case of isavuconazole and amphotericin B. On the other hand, indifference, additivity, and synergy did not show any significance. Speaking of species-specific details, *A. flavus* did not report additivity, while *A. niger* and *A. terreus* did not show antagonism cases. Additionally, *A. terreus* had a lot of indifference cases (75%), and *A. niger* reached the highest additivity rate (54.5%). Interestingly, synergy percentages were similar for all the tested species. Statistical significance emerged in *A. flavus* (*p* < 0.05), whose identification correlated with antagonism and indifference episodes. Furthermore, *A. niger* reported statistical significance (*p* < 0.05), mainly corresponding to additivity episodes.

## 3. Discussion

Invasive pulmonary aspergillosis is currently considered to be a fatal opportunistic infection, especially among critically ill patients. Unfortunately, the real IPA incidence is underestimated due to diagnostic complications and difficulties in distinguishing between colonization and infection episodes [14,15,16,17,18,19]. Scientific evidence has demonstrated that most IPA cases correspond to probable invasive aspergillosis, reporting a significant number of proven IPA only after post-mortem confirmations [19,20]. Despite triazoles’ first-line role in IPA treatment, azole resistance is increasing among *Aspergillus* spp. isolates. Consequently, echinocandins and amphotericin B have become alternative therapies or salvage therapies in specific cases. In vitro and in vivo studies had already furnished some proof about the importance of combining antifungal agents in IPA treatments. Those studies described data about azoles and echinocandins or azoles and amphotericin B combinations [13,14,15,16,17,18,19,20,21].

On these premises, we decided to focus our attention on in vitro synergy testing to extend the scientific literature about antifungal combinations on *Aspergillus* spp., studying species-specific response to the tested synergies. Our analysis highlighted the diffusion of four main *Aspergillus* spp., *A. fumigatus*, *A. flavus*, *A. niger*, and *A. terreus.* This information confirmed previously published epidemiological data about our geographical area [13,22,23]. The identified species diffused within specific hospital settings, reporting significant isolation rates among critically ill patients. Intensive care and hematological patients documented the highest *Aspergillus* spp. isolation percentages, probably due to their peculiar risk factors (corticosteroids’ usage, severe and prolonged neutropenia, mechanical ventilation, simultaneous respiratory viral or bacterial infections). All the tested isolates reported low MIC values of isavuconazole, underlining the importance of integrating this antifungal drug within susceptibility testing [24]. Moreover, isavuconazole has a compliant pharmacological profile due to there being no need to plan therapeutic drug monitoring (TDM), unlike with other azoles [25].

Regarding amphotericin B, the isolates reporting MIC values above the ECOFFs are increasing among all the main *Aspergillus* species, and previously published data already reported this trend [26]. On the one hand, most of the *A. terreus* strains exhibited an intrinsic resistance, while one isolate tested susceptible, probably due to antifungal tolerance mechanisms. Shahandashti et al. [27] documented a similar condition through in vitro studies investigating the minimum duration for killing 99% of the population (MDK99). Among our identified *Aspergillus* spp. isolates, a significant *A. terreus* percentage revealed elevated caspofungin MIC values, along with *A. fumigatus* and *A. niger* strains. Literature data already noticed a similar condition several years ago [28]. However, further analysis, including the minimum effective concentration (MEC) should be performed to confirm these preliminary susceptibility data.

Isavuconazole/amphotericin B combinations did not report any species-specific statistical significance, demonstrating concerning antagonism rates. Future in vivo confirmations may be interesting to evaluate the clinical impact of this discouraging preliminary in vitro data. On the other hand, statistical significances emerged from isavuconazole/caspofungin synergy assays, with positive results only for *A. niger*, with the promising additivity percentage. Previously published data documented possible azole/amphotericin B antagonism due to their action against the same target [29]. We justified some failures in synergy attempts by hypothesizing that azoles’ inhibition of sterols synthesis may interfere with amphotericin B target research.

Despite the similar synergy rates, isavuconazole/caspofungin revealed more in vitro effectiveness due to lower antagonism rates (14.8%) when compared to isavuconazole/amphotericin B combinations (antagonism percentage of 43.6%). Unfortunately, *A. flavus* complicated the overall deductions because of its concerning indifference and antagonism rates for both the tested combinations. On the basis of our preliminary results, antifungal combinations are not recommended for this species, while isavuconazole/caspofungin synergies may be a considerable option in the case of other species. Otherwise, *A. niger* did not report antagonism episodes and revealed a frequent additivity, suggesting the possibility of including isavuconazole/caspofungin regimens against this species. According to these assumptions, it is essential to always perform species identification to plan an effective antifungal treatment. Recent experimental studies showed the azoles’ capability to induce β-glucan accumulation and fungal membrane invaginations [30]. Echinocandins inhibit β-glucan synthesis, probably interfering with these events, leading us to hypothesize about synergy failures.

Despite the importance of observing percentages and statistical significance, we further analyzed some details about the MIC values for the included *Aspergillus* spp. strains. Specifically, the three *A. fumigatus* (13.6%) reporting elevated amphotericin B MIC values revealed a significant diminution after the isavuconazole addition during the synergy assay. Two isolates showed synergy (66.6%), while one showed additivity (33.3%) for the isavuconazole/amphotericin B combination. Conversely, the *A. fumigatus* strain showing a high caspofungin MIC value (4.5%) did not reveal advantages from the isavuconazole addition: the caspofungin MIC value decreased, but the assay resulted in indifference. These data suggest that isavuconazole/amphotericin B combinations may be effective in the case of *A. fumigatus* with amphotericin B high MIC values. One *A. niger* isolate with high amphotericin B MIC (9.1%) resulted in indifference for the isavuconazole/amphotericin B combination. Otherwise, the strain with a caspofungin high MIC value (9.1%) reported synergy for the isavuconazole/caspofungin association.

*A. flavus* did not reveal high caspofungin MIC values, but one strain (7.4%) exhibited an amphotericin-elevated MIC value. This strain had an antagonism result in the case of the isavuconazole/amphotericin B combination, enforcing the hypothesis to discourage antifungal combinations for this species. Regarding the seven *A. terreus* isolates reporting high amphotericin B MIC values (87.5%), the isavuconazole/amphotericin B combination did not often improve the initial condition, resulting in four cases (57.1%) of indifference, three cases (28.5%) of antagonism, and only one case (14.3%) of synergy. Five *A. terreus* isolates revealed elevated caspofungin MIC values (62.5%), demonstrating only indifference episodes for the isavuconazole/caspofungin combination. These data led us to hypothesize the use of single isavuconazole regimens in the case of *A. terreus*, due to elevated amphotericin B and caspofungin MIC values not always being related to successful synergy results.

Certainly, our experimental evaluation had some limitations. The retrospective analysis included a few isolates’ numbers for some species, especially regarding *A. niger* and *A. terreus.* This condition probably affected all the statistical considerations. It would be ideal to perform similar investigations, including the same number for all the tested species, but extending the sample size. Further studies should test the in vitro synergy effect by inserting antifungal molecules during different time intervals to test their effectiveness in the case of subsequent and not simultaneous action against the fungal strain.

We hypothesized this future perspective according to the frequent antifungal sequential therapy usage in the case of invasive fungal infections. Finally, advanced molecular protocols may be applied to further investigate *Aspergillus* spp. resistance patterns, aiming to explain the detected differences in synergy results within the same species.

## 4. Materials and Methods

### 4.1. General Information and Isolates Collection

We performed a retrospective analysis at the University Hospital Policlinico of Catania, encompassing the Laboratory Analysis Unit during a two-year (2022–2024) period. The study included *Aspergillus* spp. isolates from respiratory samples (bronchial aspirates, sputum samples, and bronchoalveolar lavage fluids). These samples were derived from intensive care, infectious disease, pneumology, internal medicine, surgery, emergency room, oncology, and transplant units. The study retrospectively classified patients with probable invasive pulmonary aspergillosis according to the European Organization for the Research and Treatment of Cancer (EORTC) [20,31,32]. Consequently, IPA cases derived from the presence of at least one host risk factor (neutropenia, transplants, immunosuppression), one clinical criterion (halo sign, air crescent sign, dense well-circumscribed lesions, lobar consolidation), and mycological evidence (positive microscopical and/or culture assays, galactomannan antigen detected in plasma or serum or bronchoalveolar lavage with the minimum index value of 1, duplicate positive molecular results on serum or plasma or bronchoalveolar lavage or whole blood). Cases documenting probable COVID-19-associated pulmonary aspergillosis (CAPA) were similarly categorized according to the specific CAPA diagnostic guidelines [33]. None of the gathered results had a clinical or therapeutic application. Furthermore, all the collected data were anonymized, and no patient-identifiable data was included within the study. According to these assumptions, ethical approvals were not mandatory for our local legislation. All the isolates were stored at −80 °C in sterile water after the routine diagnostic procedures. Consequently, they were restored and inoculated into R.P.M.I agar (Liofilchem^®^ s.r.l., Roseto degli Abruzzi, Italy) to perform the experimental study.

### 4.2. Isolates Identification and Susceptibility Testing

The MALDI Biotyper^®^ Sirius System (Bruker, Billerica, MA, USA) identified the grown filamentous fungi colonies according to the MBT Filamentous Fungi IVD Library Version 2022. The above-mentioned library allowed to sensu strictu (s.s.) *Aspergillus* isolates identification. The identified strains were subjected to isavuconazole, amphotericin B, and caspofungin susceptibility testing through the MIC Strip method and the gradient test principle [34,35]. The obtained MIC values were confirmed through the EUCAST broth microdilution guidelines on filamentous fungi [36]. We considered susceptible all the isolates reporting MIC values under or equal to the CBP, along with the isolates showing MIC values under the ECOFF. On the other hand, we classified resistant all the isolates showing MIC values over the CBP, along with the ones reporting MIC values under or equal to the ECOFF [37,38].

### 4.3. Synergy Assays

Isavuconazole/amphotericin B and isavuconazole/caspofungin were arranged to study synergy, following the manufacturer’s instructions [39].

The same instructions were established to perform all the single and combination assays on R.P.M.I. agar (Liofilchem^®^ s.r.l., Roseto degli Abruzzi, Italy), incubating the inoculated plates for 24–48 h at 37 °C. Finally, the synergy results were managed by calculating the corresponding FIC index, which allowed for the categorization of the results as antagonism, indifference, synergy, or additivity cases [36]. Specifically, a FIC index equal to or lower than 0.5 indicated synergy, while antagonism emerged from a FIC index higher than 4. A FIC index higher than 0.5 but equal to or lower than 1 indicated additivity, whereas a value higher than 1 but equal to or lower than 4 was related to indifference. Antagonism reported a combination with less effect than the single antifungals. Indifference is derived from the total absence of interactions between the two molecules. Additivity occurred when the combination demonstrated a higher effect than the single compounds. Synergy emerged when the combination had a more intense effect than the simple addition of the two drugs [40].

### 4.4. Statistical Analysis

The MedCalc Statistical Software version 17.9.2 “http://www.medcalc.org” (Accessed on 28 August 2025) was used to perform a statistical analysis of Aspergillus species and synergy result correspondence. Moreover, the χ2 and Fisher’s exact test established the categorical variables as percentages. Our experimental protocol did not involve any direct action on human beings. The susceptibility analyses were related to biological samples and fungal isolates, which were not collected as supplementary specimens and derived from previous routine diagnostic procedures.

## 5. Conclusions

Invasive pulmonary aspergillosis treatment poses a significant clinical challenge. Certainly, caspofungin exhibits limited activity against *Aspergillus* spp. due to its fungistatic attitude. On the other hand, amphotericin B and isavuconazole extensively demonstrated efficacy against the same molds [3,4,5,6]. However, amphotericin B recently demonstrated an increase in potential antifungal resistance rates [41]. Isavuconazole reported encouraging in vitro and in vivo results against *Aspergillus* spp. [25,42], always showing low MIC values. Despite the concerning increase in aspergillosis treatment issues, scientific literature does not sufficiently document in vitro synergy studies. On this premise, in vivo antifungal combinations are often discouraged, excepting for specific cases such as fluconazole and amphotericin B usage in the case of cryptococcosis [43]. The main purpose of our study was to report microbiological susceptibility results regarding *Aspergillus* spp. clinical isolates.

Our results enhanced the importance of always performing identification and antifungal susceptibility assays for invasive molds isolates. Synergy testing demonstrated a significant species-specific trend, mainly underlining isavuconazole/amphotericin B in vitro effectiveness against *A. fumigatus* and the elevated isavuconazole/caspofungin antagonism rate for *A. flavus*. Future studies should be focused on *Aspergillus* spp. isolates and antifungal in vitro synergy testing, aiming to discourage or recommend any specific antifungal combinations depending on the isolated species.

## Figures and Tables

**Figure 1 antibiotics-14-00993-f001:**
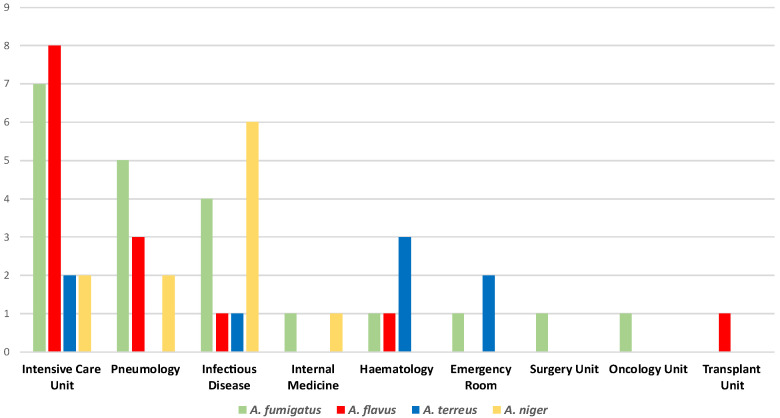
Distribution of the *Aspergillus* spp. isolates within the included hospital units.

**Table 1 antibiotics-14-00993-t001:** MIC values of amphotericin B, caspofungin, and isavuconazole for all the tested isolates, along with hospital wards, isolated species, and biological samples.

*Species*	Ward	Sample	IVU MIC (mg/L)	AMB MIC (mg/L)	CAS MIC (mg/L)
*A. fumigatus*	Internal Medicine	BAL	0.125	0.19	0.19
*A. fumigatus*	Intensive Care	BAS	0.19	0.19	0.19
*A. fumigatus*	Infectious Diseases	BAL	0.125	0.19	0.25
*A. fumigatus*	Intensive Care	BAL	0.125	0.25	0.094
*A. fumigatus*	Intensive Care	BAL	0.125	0.25	0.25
*A. fumigatus*	Surgery Unit	BAL	0.064	0.25	0.19
*A. fumigatus*	Pneumology	Sputum	0.19	0.25	0.38
*A. fumigatus*	Infectious Diseases	BAL	0.094	0.25	0.5
*A. fumigatus*	Intensive Care	BAL	0.19	0.25	0.19
*A. fumigatus*	Pneumology	BAL	0.125	0.25	64
*A. fumigatus*	Infectious Diseases	BAL	0.19	0.38	0.064
*A. fumigatus*	Intensive Care	BAL	0.25	0.38	0.19
*A. fumigatus*	Intensive Care	BAL	0.19	0.38	0.19
*A. fumigatus*	Intensive Care	BAS	0.19	0.38	0.094
*A. fumigatus*	Infectious Diseases	Sputum	0.19	0.38	0.25
*A. fumigatus*	Pneumology	BAS	0.19	0.5	0.094
*A. fumigatus*	Pneumology	Sputum	0.38	0.5	0.125
*A. fumigatus*	Pneumology	BAL	0.19	0.75	0.064
*A. fumigatus*	Oncology	BAL	0.19	0.75	0.19
*A. fumigatus*	Intensive Care	BAS	0.38	2	0.064
*A. fumigatus*	Emergency Room	BAL	0.19	4	0.19
*A. fumigatus*	Hematology	Sputum	0.38	>32	0.125
*A. flavus*	Pneumology	Sputum	0.094	0.002	0.25
*A. flavus*	Transplants unit	Sputum	0.047	0.002	0.19
*A. flavus*	Intensive Care	BAS	0.012	0.06	0.25
*A. flavus*	Intensive Care	Sputum	0.047	0.38	0.125
*A. flavus*	Intensive Care	BAL	0.25	0.75	0.38
*A. flavus*	Intensive Care	BAS	0.25	1.5	0.25
*A. flavus*	Intensive Care	BAS	0.25	8	0.064
*A. flavus*	Intensive Care	BAS	0.38	1	0.38
*A. flavus*	Pneumology	BAL	0.19	0.75	0.25
*A. flavus*	Pneumology	Sputum	0.19	0.5	0.125
*A. flavus*	Infectious Diseases	BAS	0.25	0.5	0.25
*A. flavus*	Intensive Care	BAS	0.25	0.75	0.25
*A. flavus*	Intensive Care	BAS	0.25	0.75	0.094
*A. flavus*	Hematology	BAS	0.125	0.064	0.125
*A. niger*	Internal Medicine	BAL	0.5	0.047	64
*A. niger*	Infectious Diseases	BAL	0.19	0.25	0.064
*A. niger*	Infectious Diseases	BAL	0.19	0.25	0.064
*A. niger*	Infectious Diseases	BAS	0.19	0.25	0.064
*A. niger*	Pneumology	BAL	0.19	0.25	0.19
*A. niger*	Pneumology	BAL	0.19	0.25	0.25
*A. niger*	Intensive Care	BAL	0.19	0.25	0.25
*A. niger*	Infectious Diseases	BAL	0.5	0.25	0.5
*A. niger*	Infectious Diseases	BAS	0.5	0.25	0.5
*A. niger*	Intensive Care	BAL	0.19	0.38	0.25
*A. niger*	Infectious Diseases	BAL	0.19	8	0.064
*A. terreus*	Infectious Diseases	BAL	0.5	0.047	64
*A. terreus*	Hematology	Sputum	0.064	0.38	64
*A. terreus*	Hematology	Sputum	0.064	4	64
*A. terreus*	Hematology	BAS	0.064	4	64
*A. terreus*	Emergency Room	BAS	0.064	32	64
*A. terreus*	Intensive Care	BAL	0.125	32	0.19
*A. terreus*	Emergency Room	BAL	0.25	32	64
*A. terreus*	Intensive Care	BAS	0.25	32	0.19

Abbreviations: IVU = isavuconazole; AMB = Amphotericin B; CAS = caspofungin; MIC = minimum inhibitory concentration; BAL = bronchoalveolar lavage fluid; BAS = bronchial aspirate.

**Table 2 antibiotics-14-00993-t002:** MIC ranges of amphotericin B, caspofungin, and isavuconazole for all the tested isolates, along with percentages of probable resistant or resistant isolates according to ECOFF and BP interpretation.

*Aspergillus* spp.	MIC Range (mg/L)	% >ECOFF/BP	S/R	MIC90	ECOFF/BP/HC
***A. fumigatus* (22)**					
Amphotericin B	0.125–64	13.6	19/3	2	1
Caspofungin **	0.064–64	4.5	21/1	0.38	32
Isavuconazole	0.064–0.38	0	22/0	0.38	2
***A. flavus* (14)**					
Amphotericin B *	0.002–8	7.1	13/1	1.5	4
Caspofungin **	0.064–0.38	0	14/0	0.38	32
Isavuconazole	0.012–0.38	0	14/0	0.25	2
***A. niger* (11)**					
Amphotericin B	0.047–0.38	9.1	10/1	0.38	1
Caspofungin **	0.064–64	9.1	10/1	0.5	32
Isavuconazole *	0.19–0.5	0	11/0	0.5	4
***A. terreus* (8)**					
Amphotericin B *	0.094–64	87.5	1/7	32	8
Caspofungin **	0.19–64	62.5	3/5	64	32
Isavuconazole	0.064–0.38	0	8/0	0.25	1
**TOTAL (55)**					

Abbreviations: MIC = minimum inhibitory concentration; ECOFF = epidemiological cut-off; BP = clinical breakpoint; HC = highest tested concentration; S/R = Susceptible/Resistant. * In the absence of clinical breakpoints, all the MIC values higher than the ECOFF have been evaluated as probable resistance cases; ** In the absence of both clinical breakpoint and epidemiological cut-off, all the MIC equal or higher than the latest tested antifungal dilution have been evaluated as probable resistance cases.

**Table 3 antibiotics-14-00993-t003:** In vitro synergy testing results regarding isavuconazole and amphotericin B for all the included isolates.

*Aspergillus* Species	Additivity (%)	Indifference (%)	Antagonism (%)	Synergy (%)	*p* Value
*A. fumigatus* (22)	6 (27.7)	8 (36,4)	6 (27.7)	2 (9.1)	0.218
*A. flavus* (14)	1 (7.14)	3 (21.4)	10 (71.4)	0	0.089
*A. niger* (11)	2 (18.2)	2 (18.2)	6 (54.5)	1 (9.1)	0.763
*A. terreus* (8)	1 (12.5)	4 (50.0)	2 (25.0)	1 (12.5)	0.499
**Total (55)**	**10 (18.2)**	**17 (30.9)**	**24 (43.6)**	**4 (7.8)**	

**Table 4 antibiotics-14-00993-t004:** In vitro synergy testing results regarding isavuconazole and caspofungin for all the included isolates.

*Aspergillus* Species	Additivity (%)	Indifference (%)	Antagonism (%)	Synergy (%)	*p* Value
*A. fumigatus* (22)	8 (36.4)	10 (45.4)	3 (13.6)	1 (4.5)	0.729
*A. flavus* (14)	0	7 (50)	5 (35.7)	1 (7.1)	0.011
*A. niger* (11)	6 (54.5)	1 (9.1)	0	1 (9.2)	0.012
*A. terreus* (8)	1 (12.5)	6 (75)	0	1 (12.5)	0.235
**Total (55)**	**15 (27.3)**	**24 (43.6)**	**8 (14.8)**	**4 (7.8)**	

## Data Availability

All the gathered results are included in the present manuscript.

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
