# Peer review of "Isavuconazole-Amphotericin B and Isavuconazole-Caspofungin In Vitro Synergic Activity Against Invasive Pulmonary Aspergillosis Molds Isolates"

_antibiotics, 2025, doi:10.3390/antibiotics14100993_

Round 1
Reviewer 1 Report
Comments and Suggestions for Authors
Major Comments
Lines 48–52
The statement regarding CYP51A mutations and the overexpression of transporters is correct; however, it is not supported by citations. Appropriate references should be added.
Lines 107–108
The authors need to clarify whether their isolates are A. fumigatus, A. flavus, A. terreus, etc. sensu stricto, or whether they belong more broadly to sections such as Fumigati, Nigri, or Terrei. This distinction is critical for interpreting antifungal susceptibility data.
Table 1
It would add significant value, given the scarcity of such information in the literature, to verify whether the amphotericin B–resistant A. fumigatus isolates are truly A. fumigatus sensu stricto or cryptic species (e.g., A. lentulus). The same clarification is recommended for the amphotericin B–resistant A. flavus.
Tables 2 and 3
While the authors tested isavuconazole + amphotericin B and isavuconazole + caspofungin, it would have been logical and informative to also test amphotericin B + caspofungin.
Lines 216–229 (Discussion)
The discussion could be significantly enriched by incorporating literature on the lack of synergy between azoles and amphotericin B, as well as between azoles and echinocandins. For instance, studies have shown that azole treatment can cause β-glucan accumulation in the periplasmic space, leading to membrane invaginations and rupture, suggesting a secondary antifungal mechanism beyond ergosterol inhibition. Inhibiting β-glucan synthesis with echinocandins could interfere with this mechanism, explaining the lack of synergy. Similarly, reduced ergosterol synthesis under azole treatment would logically diminish amphotericin B efficacy, as amphotericin binds to sterol-rich membrane domains.
Lines 267–318 (Materials and Methods)
This section requires clearer organization. It is recommended to divide it into subsections for each method used (e.g., E-test strips vs. broth microdilution). Presenting methods in a single block without subdivisions reduces readability. Furthermore, the first paragraph may fit better in the Introduction. A supplementary table including the MICs of individual isolates, along with their clinical origin (e.g., ICU, type of infection, patient background), would substantially enhance the transparency of the study.
Lines 320–330 (Conclusions)
The authors could emphasize that, given their data and the absence of synergy tests, the prescription of two antifungals simultaneously may not be advisable. Although rarely practiced, exceptions exist (e.g., amphotericin B combined with fluconazole in cryptococcosis), but typically sequential therapy is used instead.
Minor Comments
Lines 84–91
There is a duplicated paragraph that should be removed to improve clarity.
Missing Data Presentation
Beyond the summary table (Table 1), an additional table in the supplementary material containing the MIC results for each isolate and each method employed would be highly beneficial for readers.
General Assessment
Although the manuscript is relatively short and primarily negative in terms of results—confirming that antifungal combinations rarely demonstrate synergy—its contribution remains relevant, as it expands the dataset with additional Aspergillus isolates. The inclusion of additional tables and detailed data would considerably strengthen the manuscript and enhance its scientific value.
Author Response
Major Comments
- Comment: Lines 48–52. The statement regarding CYP51A mutations and the overexpression of transporters is correct; however, it is not supported by citations. Appropriate references should be added.
Answer: the citation has been added.
- Comment: Lines 107–108. The authors need to clarify whether their isolates are fumigatus, A. flavus, A. terreus, etc. sensu stricto, or whether they belong more broadly to sections such as fumigati, nigri, or terrei. This distinction is critical for interpreting antifungal susceptibility data.
Answer: We identified all the isolates through mass-spectrometry technologies, according to the latest version of the filamentous fungi species library. Consequently, all the identifications regarded Aspergillus fumigatus, Aspergillus terreus, Aspergillus flavus, and Aspergillus niger sensu strictu. We defined some details within the materials and methods section.
- Comment: Table 1. It would add significant value, given the scarcity of such information in the literature, to verify whether the amphotericin B–resistant A. fumigatus isolates are truly A. fumigatus sensu stricto or cryptic species (e.g., A. lentulus). The same clarification is recommended for the amphotericin B–resistant A. flavus.
Answer: We identified all the isolates through mass-spectrometry technologies. These technologies are able to distinguish Aspergillus fumigatus and Aspergillus lentulus, attributing them separated and different mass spectra. The same considerations may be applied to the other species, which were all identified by the same technologies. We revised materials and methods along with the results, adding the “sensu strictu” expression.
- Comment: Tables 2 and 3. While the authors tested isavuconazole + amphotericin B and isavuconazole + caspofungin, it would have been logical and informative to also test amphotericin B + caspofungin.
Answer: We are planning to test amphotericin B and caspofungin in vitro synergy during next studies. Thank you for this precious observation.
- Comment: Lines 216–229 (Discussion). The discussion could be significantly enriched by incorporating literature on the lack of synergy between azoles and amphotericin B, as well as between azoles and echinocandins. For instance, studies have shown that azole treatment can cause β-glucan accumulation in the periplasmic space, leading to membrane invaginations and rupture, suggesting a secondary antifungal mechanism beyond ergosterol inhibition. Inhibiting β-glucan synthesis with echinocandins could interfere with this mechanism, explaining the lack of synergy. Similarly, reduced ergosterol synthesis under azole treatment would logically diminish amphotericin B efficacy, as amphotericin binds to sterol-rich membrane domains.
Answer: Thank you for the fundamental opportunity to enrich our hypotheses. We modified the discussion, deciding to add some references and considerations according to your precious suggestion.
- Comment: Lines 267–318 (Materials and Methods). This section requires clearer organization. It is recommended to divide it into subsections for each method used (e.g., E-test strips vs. broth microdilution). Presenting methods in a single block without subdivisions reduces readability. Furthermore, the first paragraph may fit better in the Introduction. A supplementary table including the MICs of individual isolates, along with their clinical origin (e.g., ICU, type of infection, patient background), would substantially enhance the transparency of the study.
Answer: We divided the materials and methods paragraph into different sub-sections. Furthermore, we added a supplementary table including MIC values, hospital unit, and biological samples. All the patients had a CAPA diagnosis as specified at the beginning of the paragraph; thus, we did not include any further infection type information within the table.
- Comment: Lines 320–330 (Conclusions). The authors could emphasize that, given their data and the absence of synergy tests, the prescription of two antifungals simultaneously may not be advisable. Although rarely practiced, exceptions exist (e.g., amphotericin B combined with fluconazole in cryptococcosis), but typically sequential therapy is used instead.
Answer: We wish to thank you for these observations. We decided to add some sentences within the conclusions.
Minor Comments
- Comment: Lines 84–91. There is a duplicated paragraph that should be removed to improve clarity.
Answer: Sorry for this error. We remove all the repetitions.
- Comment: Missing Data Presentation. Beyond the summary table (Table 1), an additional table in the supplementary material containing the MIC results for each isolate and each method employed would be highly beneficial for readers.
Answer: We added MIC values within a new table.
- Comment: General Assessment. Although the manuscript is relatively short and primarily negative in terms of results—confirming that antifungal combinations rarely demonstrate synergy—its contribution remains relevant, as it expands the dataset with additional Aspergillus isolates. The inclusion of additional tables and detailed data would considerably strengthen the manuscript and enhance its scientific value.
Answer: We added MIC values within a new table. Thank you for all the corrections.
Please, find all the corrections highlighted in yellow.
Reviewer 2 Report
Comments and Suggestions for Authors
Dear Authors:
thank you very much for your work.
Interesting topic well developed.
Minor suggestions are in the pdf file.
Tables can be improved; I find them a little confusing, as there is so little information. You should supplement them with additional data: MIC90, as well as the range. Which ones have CB, which ones ECV, what are the cut-offs, etc.
kind regards

Author Response
Comment: Dear Authors:
thank you very much for your work.
Interesting topic well developed.
Minor suggestions are in the pdf file.
Tables can be improved; I find them a little confusing, as there is so little information. You should supplement them with additional data: MIC90, as well as the range. Which ones have CB, which ones ECV, what are the cut-offs, etc.
kind regards
Answer: Thank you for the observations. Please, find the requested integrations for the tables within the manuscript and the minor comments highlighted in yellow after their correction along the text.
Reviewer 3 Report
Comments and Suggestions for Authors
- It is required to improve introduction part with the clear explanation of the work purpose
- In the study 8 A. terreus and 11 A. niger isolates were the objects of the work. The small numbers of the strains limited statistical work.
- The isavuconazole/amphotericin B combination showed high antagonism (43.6%) and the clinical implications of this finding are not fully discussed.
- Materials methods section is very short. The description of the FIC index calculation is brief.
- The authors informed that ethical approval was not mandatory, but it would be better to state explicitly that the isolates were anonymized, and no patient-identifiable data were used.
-
The Figure 1 is not clear and could be improved by using clearer color coding for hospital units.
-
It will be better to improve table 2 and 3 with the adding percentages of the absolute counts in order to make the results easier to interpret.
- Ensure that all references are formatted according to the rules.
Author Response
- Comment: It is required to improve introduction part with the clear explanation of the work purpose.
Answer: The study purposes have been revised within the introduction section. - Comment: In the study 8 A. terreus and 11 A. niger isolates were the objects of the work. The small numbers of the strains limited statistical work.
Answer: Thank you for the observation. We specified these conditions within the discussion section, describing all the study limitations. - Comment: The isavuconazole/amphotericin B combination showed high antagonism (43.6%) and the clinical implications of this finding are not fully discussed.
Answer: Clinical implications are not fully discussed because we did not prove our findings on patients. Our considerations emerged from in vitro evaluations, which need further therapeutical and clinical confirmations before to fully discuss the implications. However, we added a sentence to explain this detail within the discussion. - Comment: Materials methods section is very short. The description of the FIC index calculation is brief.
Answer: The section has been divided into sub-sections according to the other reviewers, which required them to be short. However, we added some sentences about FIC index values according to the manufacturer’s instructions, which have been correctly cited within references’ list. - Comment: The authors informed that ethical approval was not mandatory, but it would be better to state explicitly that the isolates were anonymized, and no patient-identifiable data were used.
Answer: Thank you for this important observation. We added some sentences within the first materials and methods subsection. - Comment: The Figure 1 is not clear and could be improved by using clearer colours coding for hospital units.
Answer: We modified figure 1, making brighter some colours. - Comment: It will be better to improve table 2 and 3 with the adding percentages of the absolute counts in order to make the results easier to interpret.
Answer: You can find the absolute number of each result type before the parentheses indicating percentages. - Comment: Ensure that all references are formatted according to the rules.
Answer: They have been checked.
Round 2
Reviewer 1 Report
Comments and Suggestions for Authors
I am satisfied with responses to comments and modifications made to the manuscript.
Reviewer 3 Report
Comments and Suggestions for Authors
Manuscript can be accepted.